# Robust and accurate estimation of paralog-specific copy number for duplicated genes using whole-genome sequencing

Timofey Prodanov [1] & Vikas Bansal [2✉]

The human genome contains hundreds of low-copy repeats (LCRs) that are challenging to analyze using short-read sequencing technologies due to extensive copy number variation and ambiguity in read mapping. Copy number and sequence variants in more than 150 duplicated genes that overlap LCRs have been implicated in monogenic and complex human diseases. We describe a computational tool, Parascopy, for estimating the aggregate and paralog-specific copy number of duplicated genes using whole-genome sequencing (WGS). Parascopy is an efficient method that jointly analyzes reads mapped to different repeat copies without the need for global realignment. It leverages multiple samples to mitigate sequencing bias and to identify reliable paralogous sequence variants (PSVs) that differentiate repeat copies. Analysis of WGS data for 2504 individuals from diverse populations showed that Parascopy is robust to sequencing bias, has higher accuracy compared to existing methods and enables prioritization of pathogenic copy number changes in duplicated genes.

[1] Bioinformatics and Systems Biology Graduate Program, University of California, La Jolla, San Diego, CA 92093, USA. [2] Department of Pediatrics, School of Medicine, University of California, La Jolla, San Diego, CA 92093, USA. ✉email: vibansal@ucsd.edu

Whole-genome sequencing (WGS) has the potential to profile all genetic variants simultaneously in a genome, however, the presence of repetitive sequences in the human genome hinders the ability to achieve this potential. Segmental duplications or low-copy repeats (LCRs) are long segments of repetitive DNA that constitute 5–8% of the human genome[1,2]. More than 900 genes are known to overlap these segmental duplications and mutations in several such genes are associated with rare and complex human diseases[3]. Genes that overlap segmental duplication or have high sequence homology to other loci in the genome are problematic for short-read sequencing technologies since the reads derived from such genes have ambiguity in their alignment and are difficult to correctly position in the genome[3–5]. As a result, variants such as SNVs and short indels are difficult to identify in these genes using short reads[6].

Low-copy repeats are also highly susceptible to copy number changes including deletions and duplications as well as reciprocal crossover (gene conversion) events that can change paralog-specific copy number. Many of these copy number changes are known to be disease associated[7–11]. For example, copy number of the *SMN1/2* gene can modify phenotype for spinal muscular atrophy (SMA) and copy number changes at the *STRC* locus are known to cause hearing loss[9]. In spite of their relevance for human disease, most duplicated genes are excluded from standard WGS analysis pipelines since the presence of paralogous sequences with high sequence identity and extensive copy number variation makes it difficult to analyze these loci accurately.

To enable the detection of clinically relevant copy number variants in disease-associated duplicated genes, specialized diagnostic assays have been developed that utilize Quantitative real-time PCR (qPCR), paralog ratio tests[12,13] (PRT) and multiplex ligation-dependent probe amplification[14] (MLPA). Both qPCR and PRT utilize PCR product specificity to distinguish paralogous copies of a gene. However, these methods are labor-intensive and require the design and testing of multiple primers for each locus. Therefore, these methods cannot scale easily for copy number analysis of the hundreds of duplicated genes in the human genome. Array-based methods such as CGH can scale for multiple genes but cannot provide paralog-specific copy number which can be important for disease mapping. For example, at the *SMN1* locus (the two genes *SMN1* and *SMN2* only differ by 5 nucleotides), individuals with two copies of *SMN1* and one copy of *SMN2* are healthy while individuals with one copy of *SMN1* can be affected[15].

Analysis of read depth using WGS data mapped to a reference genome is a widely used approach for identifying copy number changes in the human genome. Over the last decade, a number of statistical methods have been developed for identifying CNVs from WGS and targeted sequencing experiments[16–21]. The vast majority of these methods calculate read-depth in non-overlapping windows of a fixed length across the genome and detect changes in the depth of coverage along chromosomes to identify CNVs. CNV detection from WGS has been shown to be more sensitive than array-CGH based CNV detection[22]. However, CNV detection methods for WGS data are designed to analyze genomic regions independently and either exclude genomic regions with low mappability from consideration or randomly place reads with low mapping quality[17] to avoid false positives. Therefore, such methods tend to have low accuracy for detecting copy number variation in LCRs. One exception is the GenomeSTRiP method that can detect CNVs in both unique and duplicated sequences[20].

Alkan et al.[23] developed a short read mapping algorithm, mrsFAST, that can identify multiple mapping locations for reads and used it to predict copy number in duplicated regions of the human genome. Building on this approach, Sudmant et al.[24] leveraged SUNs—paralogous sequence variants that uniquely tag a repeat copy—to estimate total copy number as well as paralog-specific copy number for all duplicated genes in the human genmome. Analysis of WGS data from the initial phases of the 1000 Genomes project showed that almost half (49%) of duplicated genes are copy number invariable while the remaining set of duplicated genes show extensive copy number variation with many copies not represented in the reference human genome[24]. Recently, Shen et al.[25] have developed a computational tool QuicK-mer2 that leverages a similar approach to estimate paralog-specific copy number.

Since WGS is now widely used in the clinical setting for disease diagnostics, there is strong interest in developing computational tools that can detect both copy number and sequence variation in disease-relevant duplicated genes with high accuracy[3]. Several methods—designed specifically for individual genes such as *SMN1*, *STRC*, *PMS2*—have been developed for this purpose[4,26,27]. For example, the SMNCopyNumberCaller tool[26] is designed to estimate the copy number of *SMN1*, *SMN2* and a partially deleted version of *SMN2* from WGS data. Similarly, a workflow for detecting variants in the duplicated region of *PMS2* has also been developed[28]. Although these tools are valuable for analyzing duplicated genes, they leverage prior knowledge about individual genes and are not directly applicable to other duplicated genes.

Copy number analysis for duplicated genes requires joint analysis of reads that are mapped to homologous repeat copies[20,26]. In this paper, we describe a probabilistic method, Parascopy, for estimating total (and paralog-specific) copy number of low copy repeats (LCRs) in the human genome. Our method leverages a homology database that stores positional information about similar sequences in the human genome as well as the positions at which the paralogous sequences differ (PSVs or paralogous sequence variants). It uses the homology database to extract relevant reads from existing alignments of WGS data. To avoid pitfalls associated with using polymorphic PSVs for differentiating repeat copies, Parascopy jointly estimates paralog-specific copy number and reference allele frequencies for each PSV using WGS data for multiple samples. This also identifies common profiles of copy number variation that can be used to analyze individual WGS datasets. We benchmark Parascopy's accuracy using experimental copy number datasets, Mendelian trio consistency analysis and concordance analysis on replicate WGS datasets.

## Results

**Overview of method**. Our method, Parascopy, is designed to estimate the aggregate copy number (*AggregateCN*) and paralog-specific copy number (*ParalogCN*) of low-copy repeats or LCRs in the human genome (Fig. 1a). Even though a large fraction of short reads cannot be mapped unambiguously due to the repetitive nature of such loci, it is feasible to analyze read depth jointly across the different copies of a low-copy repeat and estimate the aggregate number of copies. For a LCR *R*, Parascopy uses a homology table to quickly identify all other regions in the genome that share high sequence similarity or homology with *R*. The homology table—similar to a segmental duplication database—stores all pairs of sequences in the genome (with a minimum length and minimum similarity score) and is precomputed using standard alignment tools (see Methods).

Subsequently, reads from regions homologous to *R* are re-mapped to *R* and the aggregated reads are used to tabulate read depth in non-overlapping windows. A Hidden Markov Model (HMM) is used to segment *R* into regions of fixed copy number based on the read depth profiles and background read depth distributions. To account for variation in read depth across

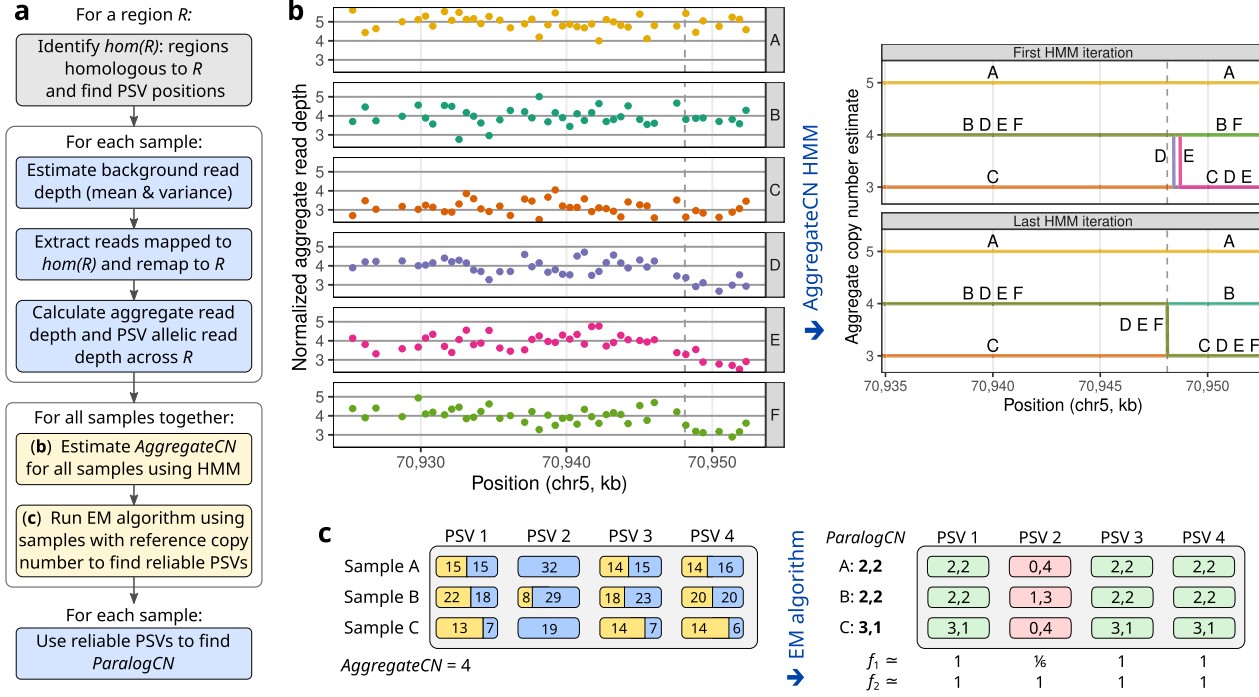

**Fig. 1 Estimation of aggregate and paralog-specific copy number for low-copy repeats using Parascopy. a** Workflow of the method using aligned WGS reads for multiple samples as input to infer aggregate and paralog-specific copy number profiles across a genomic region. **b** Illustration of the iterative Hidden Markov Model (HMM) approach for estimating aggregate copy number (*AggregateCN*) profiles using normalized read depth for multiple samples. Read depth values are shown for six samples (A–F) at the *SMN1/2* locus (aggregated across *SMN1* and *SMN2*). The HMM identifies a partial deletion in samples D and E in the first iteration. Joint update of the HMM parameters results in detection of a common deletion event in the 3 of the 6 samples. **c** Illustration of the Expectation-Maximization (EM) algorithm for estimating paralog-specific copy number (*ParalogCN*) and paralogous sequence variant (PSV) reliability. PSV reliability is measured using *f* values that correspond to the population frequency of the reference allele for each PSV at each paralogous position.

different genomic regions, the background read depth distributions are estimated for each sample and GC-content value using non-duplicated genomic regions (see Methods). The initial state distribution and transition probabilities of the HMM are estimated jointly across multiple samples enabling high sensitivity for the detection of copy number variants that are present in multiple samples (Fig. 1b).

Once the aggregate copy number profile has been estimated for each sample, Parascopy estimates the number of copies of each paralog present in the genome (*ParalogCN*) by analyzing allelic read depth at positions that differ between the homologous sequences, i.e., paralogous sequence variants or PSVs. Since some PSVs are not fixed in the population and correspond to variants, Parascopy jointly models frequency of the reference allele at each homologous position for each PSV and the *ParalogCN* for samples with *AggregateCN* equal to the reference. It considers all possible combinations of *ParalogCN* for each individual sample and uses an EM algorithm to infer maximum likelihood estimates for both sets of variables (Fig. 1c).

**Parascopy estimates copy number accurately and identifies reliable PSVs at the *SMN1/2* locus.** The *SMN1/2* locus on chromosome 5 harbors the *SMN1* gene and its paralog *SMN2* in a tandem duplication of length ≈ 100 kilobases and very high sequence identity (99.9%). Mutations—point mutations and copy number changes—in the *SMN1* and *SMN2* genes cause a rare childhood disorder called spinal muscular atrophy (SMA) and *SMN1* is one of the most-studied duplicated genes in the genome. We estimated copy number for all 2504 samples with WGS data

from phase 3 of the 1000 Genomes Project (1kGP)[29] using Parascopy (samples for each continental group were analyzed separately). Analysis of the Parascopy copy number profiles across the 1kGP samples identified a known deletion event that spans exon 7–8 (Fig. 2a) and confirmed the extensive variation in *AggregateCN* (2–6) across human populations[26].

Vijzelaar et al.[30] used MLPA to estimate *AggregateCN* of each exon of the *SMN1/2* gene for 1109 1kGP samples. For the exon 7–8 region, the copy number values for 79 of the 1109 samples were consistent with the presence of the common deletion. The *AggregateCN* estimates from Parascopy were perfectly concordant with MLPA values[30] for both exons 1–6 and 7–8 (Fig. 2b). We also compared Parascopy's accuracy for copy number estimation with three other existing methods: SMNCopyNumberCaller[26]—a method designed specifically to estimate copy number for *SMN1/2*; QuicK-mer2[25]—an alignment-free approach to estimate *ParalogCN* using *k*-mers unique to paralogous sequences; and CNVnator[17]—a CNV detection algorithm that statistically analyzes read depth from WGS data. Both SMNCopyNumberCaller and QuicK-mer2 had high accuracy for *AggregateCN* of exons 1–6 but CNVnator had a much lower accuracy equal to 68.5% (Table 1). For the exon 7–8 region, only SMNCopyNumberCaller showed high accuracy (sensitivity = 1.00 and specificity = 0.999). while both CNVnator (sensitivity = 0.823 and specificity = 0.849) and QuicK-mer2 (sensitivity = 0.709 and specificity = 0.382) had significantly lower accuracy (Supplementary Table 1). We also compared *ParalogCN* estimates from Parascopy with those from SMNCopyNumberCaller, QuicK-mer2 and CNVnator. While SMNCopyNumberCaller's estimates on 855 non-African samples were identical to Parascopy, QuicK-mer2 and CNVnator showed higher mean absolute difference of 0.53 and 0.23 respectively.

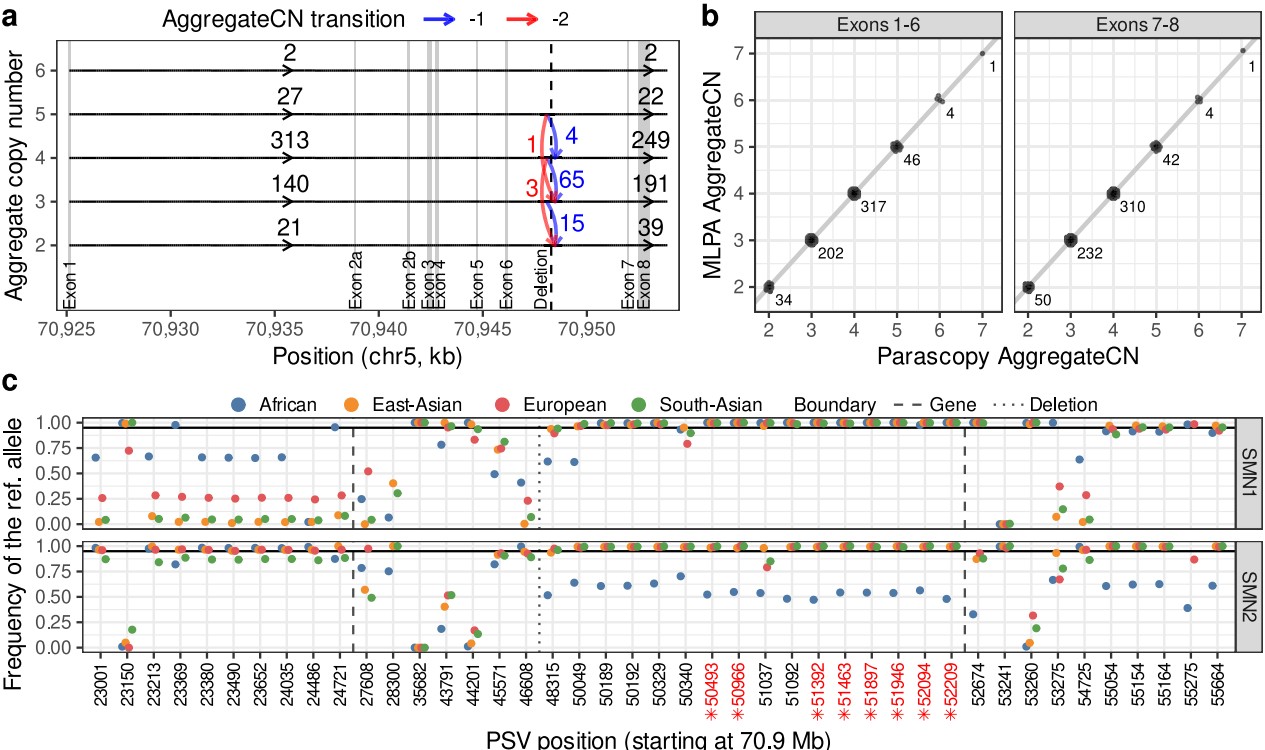

**Fig. 2 Estimation of aggregate and paralog-specific copy number for the *SMN1/2* locus using Parascopy. a** Output from the Hidden Markov Model estimation of aggregate copy number (*AggregateCN*) profiles for 503 European ancestry samples from 1kGP. The common deletion event at the 3' end of the *SMN1/2* gene is shown using blue and red arrows. **b** Comparison of the Parascopy *AggregateCN* estimates with MLPA based estimates for exons 1–6 and exons 7–8 (with deletion). Labels represent the number of samples with the corresponding copy number estimates. **c** Distribution of the frequencies of the reference alleles (*f* values) for 43 paralogous sequence variants (PSVs; 23 within *SMN1/2*) across four different 1kGP continental populations. The eight PSVs used for estimating paralog-specific copy number by SMNCopyNumberCaller are highlighted in red.

**Table 1 Accuracy of aggregate copy number estimation for three different methods across ten duplicated genes in the human genome.**

| Duplicated gene | Sample size | Copy number mean ± SD | CNVnator | QuicK-mer2 | Parascopy |
|---|---|---|---|---|---|
| *SMN1/2* | 1109 | 3.7 ± 0.6 | 68.5 (99.9) | 99.5 | 100.0[a] |
| *C4A/B* | 45 | 3.8 ± 0.6 | 86.7 | 75.6 | 100.0[a] |
| *FCGR3A/B* | 51 | 4.1 ± 0.5 | 94.1[a] | 94.1[a] | 94.1[a] |
| *PMS2/CL* | 140 | 4.0 ± 0.0 | 67.7 (92.9) | 97.9 | 100.0[a] |
| *HYDIN/2* | 5 | 4.4 ± 0.9 | 100.0[a] | 100.0[a] | 100.0[a] |
| *APOBEC3A/B* | 179 | 3.6 ± 0.6 | 94.4 | 96.1 | 96.9[a] (90.5) |
| *RHD/RHCE* | 40 | 3.6 ± 0.8 | 97.5[a] | 97.5[a] | 97.5[a] |
| *NPY4R/2* | 18 | 4.8 ± 0.8 | 66.7 | 77.8[a] | 77.8[a] |
| *SRGAP2* | 40 | 7.8 ± 0.7 | 82.5 | 62.5 | 100.0[a] |
| *AMY1A/B/C* ($\delta$) | 225 | 7.3 ± 2.6 | 0.887 (99.1) | 1.119 | 0.723[a] (96.0) |

For each method, accuracy is the percentage of samples with identical WGS-based and experimental copy number values (see Supplementary Data 2). Percentage of copy number estimates with high quality is shown in parentheses when it is below 100%. The third column in the table shows the mean and standard deviation (SD) of the experimental values. The reference copy number is 4 for all loci except for *SRGAP2* (8) and *AMY1* (6). For the *AMY1* locus, accuracy is estimated by computing mean absolute error ($\delta$) due to high variance in copy number.
Source data are provided as a Source Data file.
[a]Highest accuracy for each gene.

Unlike previous methods, Parascopy estimates the population frequency of the reference allele for each PSV (*f* values), and only uses *reliable* PSVs —PSVs with $f \geq 0.95$ for all homologous positions—to estimate *ParalogCN*. Estimates of PSV *f* values across the different populations showed that 10–19 of the 43 PSVs within and in the vicinity of *SMN1* were reliable for 4 of the 5 continental populations in the 1kGP while none of the PSVs were reliable in the African population samples (Fig. 2c and Supplementary Data 1). This was consistent with the observations of Chen et al.[26] about the lower concordance between *ParalogCN* and

values at individual PSV sites. Notably, the set of PSVs identified as reliable by Parascopy included all 8 PSVs used for estimating *ParalogCN* by SMNCopyNumberCaller[26].

**Parascopy outperforms existing methods for copy number estimation.** Next, we benchmarked the accuracy of Parascopy on additional duplicated genes with experimentally determined copy number data. For this, we compiled previously published datasets with experimental copy number data for more than 1100 samples (from the 1kGP) across nine different genes apart from *SMN1/2*

**Table 2 Accuracy of paralog-specific copy number estimates for three different methods using experimental copy number observations for four duplicated genes in the human genome.**

| Duplicated gene | Sample size | CNVnator | QuicK-mer2 | Parascopy | Reliable PSVs |
|---|---|---|---|---|---|
| SRGAP2 | 40 | 67.5 | 72.5 | 97.2[ab] | 1461/940 |
| C4A/B | 45 | 51.1 | 48.9 | 66.7[a] | 7/50 |
| FCGR3A/B | 40 | 97.5[a] | 47.5 | 97.5[a] | 120/179 |
| RHCE/RHD | 40 | 95.0 | 97.5[a] | 92.5 | 897/1027 |

The last column shows the number of reliable paralogous sequence variants (PSVs; 1kGP European samples) and the total number of PSVs within the duplicated gene or locus.
Source data are provided as a Source Data file.
[a]Highest accuracy for each gene.
[b]Paralog-specific copy number estimates have low qualities in four samples.

(listed in Supplementary Data 2). First, we compared the accuracy of AggregateCN estimates obtained from Parascopy with CNVnator and QuicK-mer2 (Table 1). Across the 9 genes, AggregateCN estimates from Parascopy were either more accurate than both methods (SRGAP2, C4A/B, PMS2, AMY1) or equally accurate (FCGR3A/B, HYDIN, APOBEC3A/B, RHD/RHCE, NPY4R/2). For the AMY1 locus, which has a high variation in total copy number (2–18) in human populations, Parascopy's mean absolute error was 0.72 compared to 0.89 and 1.12 for CNVnator and QuicK-mer2 respectively (Supplementary Fig. 1). For the APOBEC3A/B locus, Parascopy's assigned low quality (<20) to copy number values for 9.5% of samples due to the small length of the gene. The lowest accuracy (77.8% on 18 samples) for Parascopy was observed for the NPY4R/2 locus. Visual inspection of read depth profiles at this locus for the 18 samples indicated that Parascopy's estimates are likely to be correct for all samples and were perfectly concordant with QuicK-mer2 estimates (Supplementary Fig. 2).

Next, we assessed the accuracy of paralog-specific copy number estimation for the three methods across 4 of the 9 genes that had experimental paralog-specific copy number data (Table 2). Parascopy's average accuracy (87.58%) was greater than both CNVnator (76.97%) and QuicK-mer2 (66.06%). Estimation of ParalogCN depends on PSVs that can differentiate the repeat copies and all methods had low accuracy for the C4/B locus which had a low number of reliable PSVs (7/50).

Finally, we compared the performance of the different methods for identifying the boundaries of copy number changes within a gene. For this, we analyzed the PMS2/PMS2CL locus where 4 of 150 1kGP samples were reported to harbor a partial deletion covering two exons (exons 13 and 14) using LR-PCR sequencing and MLPA[28]. Analysis of the AggregateCN profiles estimated by Parascopy's HMM showed that a partial deletion was correctly identified in 4/4 samples albeit with low quality (<20) in 2 of the 4 samples (Supplementary Fig. 3). Parascopy did not identify the deletion event in any of the remaining samples (sensitivity = 1.0 and specificity = 1.0). In contrast, QuicK-mer's copy number profiles showed no evidence of the deletion (sensitivity = 0.0 and specificity = 1.0) while CNVnator detected a copy number change in 3/4 samples (sensitivity = 0.75 and specificity = 0.691).

**Accuracy of Parascopy copy number estimates across a set of genome-wide low-copy repeats.** Next, we evaluated Parascopy's accuracy and robustness for estimating copy number across a larger set of duplicated coding loci in the human genome. For this purpose, we compiled a catalog of 167 low-copy repeat loci—overlapping over 220 protein-coding genes (380 including homologous regions)—using previous analysis of sequence homology of coding regions in the human genome[3] and copy number estimates for genes overlapping segmental duplications[24] (see Methods). These 167 low-copy repeat loci span 12.6 Mb

of DNA sequence (including homologous regions) and 65.0 (14.7)% of these loci correspond to two (three) copy duplications (see Supplementary Data 3 for a complete list).

First, to assess the robustness of the copy number estimates to variation in sequencing bias, we analyzed each of the 167 repeat loci in a set of 90 individuals of Han Chinese ancestry for which WGS data was generated independently by Lan et al.[31] using a PCR-based library preparation protocol. 83 of these 90 individuals also had WGS data available from the 1kGP generated using a PCR-free library preparation protocol. In comparison with the PCR-free data, the PCR-based WGS data exhibited significant greater biases in the distribution of read depth as a function of GC-content (Supplementary Figs. 4 and 5). We ran Parascopy, CNVnator and QuicK-mer2 on the two datasets independently and compared the concordance between pairs of replicate samples (across the 167 repeat loci) for each method. Parascopy reported AggregateCN estimates (with quality ≥20) for 94.5% of the pairs and 98.7% of the AggregateCN pairs were concordant. In comparison, QuicK-mer2 provided AggregateCN values for 100% of the pairs with a concordance rate of 74.9% (Table 3). CNVnator's concordance (86.9% with a completeness of 97.0%) was also significantly lower than Parascopy. Notably, Parascopy's concordance without any quality value filter (96.4%) was still 11.4 perecentage points greater than that for CNVnator. These results also showed that Parascopy AggregateCN values with quality <20 are less reliable.

Parascopy does not estimate ParalogCN values for loci that have high reference copy number or a low fraction of reliable PSVs (see Methods). As a result, the concordance analysis was limited to the 122 loci that had ParalogCN for one or more samples across both replicates. Across these loci, Parascopy's ParalogCN had a concordance rate of 99.8% (99.5%) for a quality threshold of 20 (0). Notably, the mean absolute difference between replicates was 0.003. In comparison, QuicK-mer2 and CNVnator ParalogCN estimates were available for all loci and had a concordance rate of 81.1% and 85.5% respectively (Table 3). For the smaller set of 122 duplicated loci with ParalogCN estimates from Parascopy, QuicK-mer2 and CNVnator average concordance values were 85.4% and 93.3% respectively, higher than those for all loci. At the SMN1 locus, the PSV f values, estimated by Parascopy, were highly concordant between the two datasets ($r^2 > 0.92$) and the same set of 20 PSVs were identified as reliable in both datasets (Supplementary Fig. 6).

Next, we used trio analysis to assess if the ParalogCN values estimated by Parascopy are consistent with Mendelian rules of inheritance. For this, we utilized 602 trios with WGS data from the expanded 1kGP dataset[32]. To account the uncertainty in the locus-specific ParalogCN values for a trio, we used a probabilistic method to calculate a probability that the trio ParalogCN values are concordant with Mendelian inheritance (see Methods). We analyzed trio concordance for 137 of the 167 loci, for which Parascopy could estimate high quality ParalogCN values. On average, 99.5% trios were concordant per loci, with 126 loci having at least 99% concordant trios (Supplementary Data 4). The concordance rate for the subset of

**Table 3 Concordance of aggregate (*AggregateCN*) and paralog-specific (*ParalogCN*) copy number estimates across 167 duplicated loci between two replicate WGS datasets for 83 Han Chinese samples.**

| Data type | Metric | CNVnator | | QuicK-mer2 | Parascopy | |
|---|---|---|---|---|---|---|
| | | Q ≥0 | Q ≥20 | | Q ≥0 | Q ≥20 |
| *AggregateCN* | Available estimates (%) | 100.0 | 97.0 | 100.0 | 100.0 | 94.5 |
| (167 loci) | Concordance (%) | 85.0 | 86.9 | 74.9 | 96.4 | 98.7 |
| | Mean absolute difference | 0.185 | 0.157 | 0.292 | 0.041 | 0.014 |
| *ParalogCN* | Available estimates (%) | 100.0 | 97.0 | 100.0 | 72.8 | 70.1 |
| (167 loci) | Concordance (%) | 83.4 | 85.5 | 81.1 | 99.5 | 99.8 |
| | Mean absolute difference | 0.282 | 0.240 | 0.299 | 0.007 | 0.003 |
| *ParalogCN* | Available estimates (%) | 100.0 | 97.8 | 100.0 | 99.7 | 96.0 |
| (122 loci) | Concordance (%) | 91.4 | 93.3 | 85.4 | 99.5 | 99.8 |
| | Mean absolute difference | 0.129 | 0.101 | 0.186 | 0.007 | 0.003 |

Q ≥ 0—use all copy number estimates; Q ≥ 20—use only high quality copy number estimates. QuicK-mer2 does not have a quality measures, therefore all copy number estimates were used. 122 loci—a subset of loci where Parascopy estimates *ParalogCN* for at least one sample in both datasets.
Source data are provided as a Source Data file.

locus-trio pairs for which the predicted *ParalogCN* for the child was >2, was 95.5% (4776/5093).

Parascopy can estimate copy number values for individual samples by utilizing the model parameters (HMM parameters and PSV *f* values) inferred from an independent set of samples (see Methods). To assess the accuracy of Parascopy for individual samples, we analyzed 210 samples from two populations in the 1kGP (IBS and CHB) and compared the *AggregateCN* values for each sample obtained by individual estimation (using the model parameters from the other population) with multi-sample estimation (all samples from each population analyzed jointly). Parascopy *AggregateCN* estimates were perfectly concordant (Supplementary Table 2) for 165 of the 167 loci. The two remaining loci (*PRAMEF1* and *RHPN2*) had a mean *AggregateCN* > 7 and did not have high quality estimates available for comparison. Similarly, *ParalogCN* estimates showed very high concordance equal to 98.9%.

The accuracy of copy number estimation is expected to improve with increasing read depth. The mean read depth for the 1kGP samples was 33×. To assess the accuracy of Parascopy at lower values of sequence coverage, we sub-sampled WGS data for 107 samples from the IBS population in the 1kGP to one-third and two-thirds of the original read depth, analyzed them using model parameters from a different continental population and compared copy number estimates with those obtained using the full coverage (see Methods). As expected, the percentage of high-quality *AggregateCN* estimates reduced with decreasing read depth: 94% at two-thirds and 88.3% at one-third coverage (Supplementary Table 2). Nevertheless, the high-quality *AggregateCN* and *ParalogCN* estimates had high concordance equal to 99.9% and 98.4% respectively at one-third coverage.

Parascopy is multi-threaded and can process multiple loci in parallel. Analyzing 503 European genomes from the 1kGP took 17 h using 16 cores and required <12 Gb of memory. For a single genome with 30 × WGS, Parascopy took 16 min to analyze 167 duplicated gene loci using 16 threads and required <5 Gb of memory. In comparison, CNVnator (QuicK-mer2) took 28 (36) min to analyze a single genome using 16 threads and required 12 (40) Gb of memory. We note that a direct comparison of run-time between Parascopy, CNVnator and QuicK-mer2 is difficult since CNVnator and QuicK-mer2 are genome-wide methods while Parascopy is a targeted copy number estimation method. Nevertheless, the low memory requirements and run-time for Parascopy allow it to scale up for analyzing thousands of samples efficiently.

**Analysis of copy number changes and PSVs across 2504 individuals.** To explore the diversity of copy number at low copy repeat loci across populations and genes, we estimated copy number at the 167 repeat loci for all 2504 individuals from five continental populations sequenced in the 1kGP. For 151 of the 167 loci, *AggregateCN* values could be estimated with high confidence (quality ≥ 20) for at least 95% of the samples. High average copy number was the main reason for the low quality of the *AggregateCN* estimates at some loci. The mean *AggregateCN* was 4.39 (6.42) for the 151 (16) loci with ≥95% (<95%) of the samples with high confidence *AggregateCN* values (Supplementary Data 5). Similarly, for 26 of the 167 loci, *ParalogCN* estimates were not estimated either due to a low number of reliable PSVs (e.g., *CFC1*) or due to the lack of a sufficient number of individuals with reference copy number (see Methods).

Not surprisingly, the most frequent copy number value for the vast majority (88.4%) of loci was equal to the reference copy number. Several disease-associated genes had a low variance in aggregate copy number (e.g., *HYDIN*) while other genes such as *SMN1/2* and *NEB* had a large variance in the copy number. Among 164 loci, 84 loci had 99% or greater of samples with *AggregateCN* equal to the reference (Fig. 3b). For 15 loci, the *AggregateCN* for more than half of the samples was greater than the reference—likely due to a missing copy in the reference genome (hg38) used for analysis. Notably, the most frequent copy number for the *OTOA* gene locus (*OTOA + OTOAP1*) with a reference copy number of 4 was 6 (Fig. 4a). To investigate this further, we leveraged the recent highly complete human genome assembly from the T2T consortium for the CHM13 cell line[33]. Alignment of the *OTOA* duplicated sequence to this assembly revealed the presence an additional copy that is not present in the current human reference genome and has sequence similarity ≥99.5% to the two other copies. We added this additional copy to the reference genome and re-analyzed the 1kGP samples using Parascopy. The *AggregateCN* estimates were not affected by the presence of the additional copy (concordance = 100%) demonstrating the robustness of Parascopy's *AggregateCN* estimates in the presence of missing repeat copies. In addition, we were able to estimate *ParalogCN* values and identify reliable PSVs using the sequence information from the additional copy. Analysis of the *ParalogCN* values across the 1kGP data showed that *OTOAP1* locus is the most polymorphic in terms of copy number. For example, out of 540 samples with *AggregateCN* = 5, more than 93% samples were missing one copy of *OTOAP1*.

Next, we analyzed the frequency and distribution of copy number changes in individual disease-associated genes and their relationship with known pathogenic variants. For the *STRC* gene, ~1.5% of individuals across all continental populations were

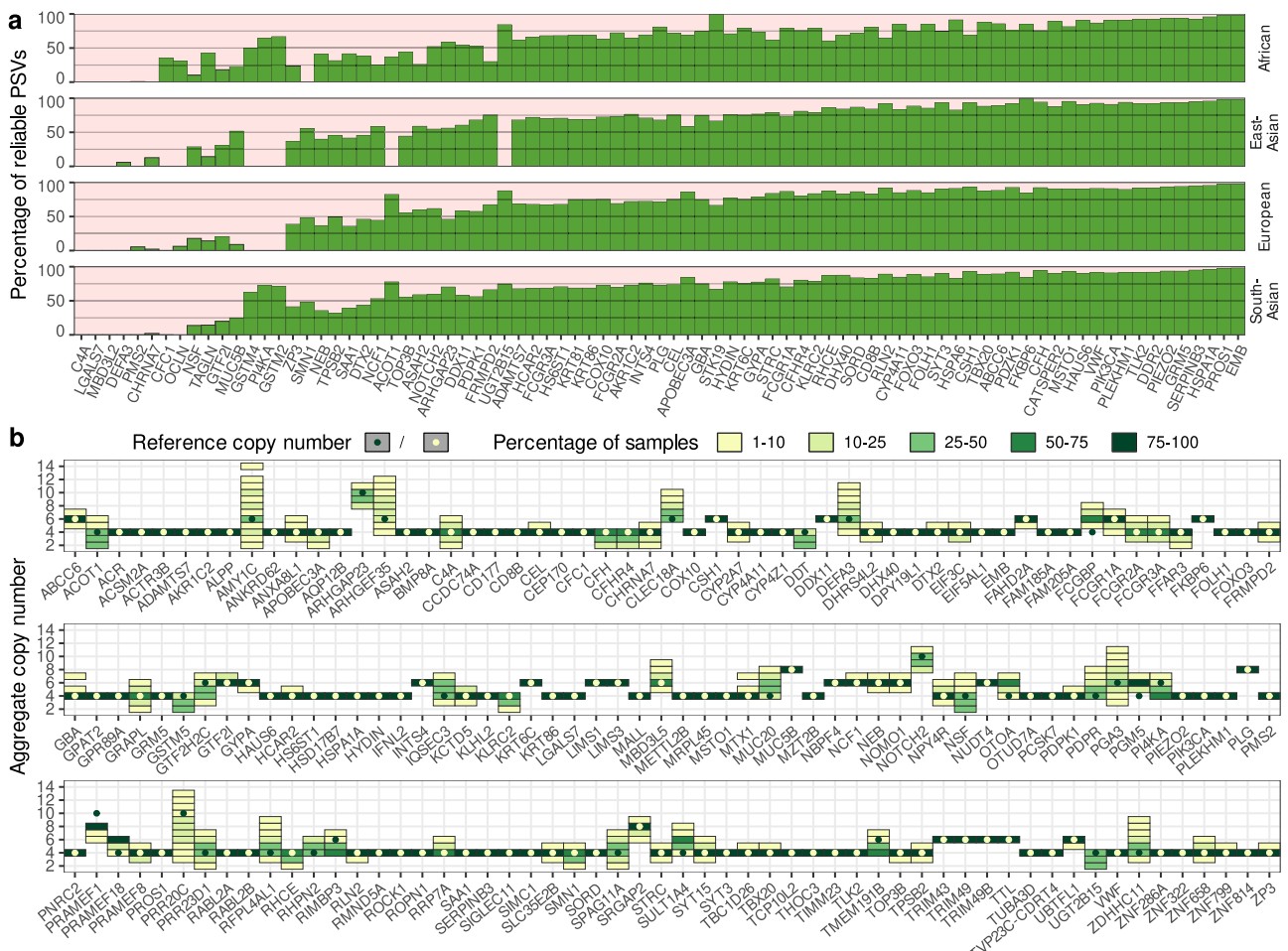

**Fig. 3 Distribution of the percentage of reliable paralogous sequence variants (PSVs) and aggregate copy number (*AggregateCN*) profiles across duplicated genes. a** Percentage of reliable PSVs ($f \geq 0.95$) across 83 disease-associated genes and four continental populations from 1kGP. **b** Distribution of *AggregateCN* for 167 duplicated loci across all populations. Dark/white dots show reference copy number for each locus. Rare events (<1% samples) are not shown.

carriers of a heterozygous deletion of *STRC* while no individual had a bi-allelic deletion (Fig. 4b). Bi-allelic deletions in this gene are known to cause hearing loss[4,34]. At the *GBA* locus—variants in *GBA* are associated with Gaucher disease[35]—we observed that 9.4% of individuals from African populations had an *AggregateCN* of 6 or more while only 2 individuals (0.1%) from non-African populations had such high copy number (Fig. 4d). *GBA* and *GBAP1* (pseudo-gene) are located in homologous repeats separated by 10 kb on chromosome 1. Further analysis of copy number revealed that the increased copy number is a result of a duplication that includes the last two exons of *GBA*, part of *GBAP1* and the entire region between the two repeats (Supplementary Fig. 7). For the *NEB* gene locus that harbors an intragenic repeat with three copies, the aggregate copy number varied from a minimum of 2 (one sample) to a maximum of 8 (population allele frequency of 0.12%, Fig. 4c). Previous analysis of *NEB* copy number in 60 controls using a custom CGH microarray[36] had indicated that copy number gains of 2–4 copies could be pathogenic for nemaline myopathy. Our results on a much larger number of population samples indicate that copy number gains of 2 copies are observed at a low frequency and are unlikely to be pathogenic. Furthermore, the observed frequency of 1-copy gains and losses (3.1% and 5.4%) were consistent with those observed using CGH data (3.9% and 5.4%).

The fraction of reliable PSVs varied significantly across genes with some well-studied disease genes such as *C4A* and *PMS2*

having a very low fraction of reliable PSVs while >90% of the PSVs were reliable for genes such as *VWF* and *ABCC6* (Fig. 3a and Supplementary Data 6). The fraction of reliable PSVs was highly correlated across populations ($r^2 = 90\%$, on average sets of reliable PSVs overlap by more than 95%) except for a few genes such as *SMN1* for which no reliable PSVs were identified for the African population. A high fraction of unreliable PSVs ($f < 0.95$) makes it challenging to estimate *ParalogCN*. Comparison of Parascopy's *ParalogCN* estimates with those estimated by QuicK-mer2 for several disease-associated genes (Supplementary Fig. 8) showed that while the *AggregateCN* estimates were highly concordant between the two methods, the concordance of the *ParalogCN* estimates was low for genes with a high frequency of unreliable PSVs. For example, the correlation coefficient $r^2$ between the *ParalogCN* values for Parascopy and QuicK-mer2 (using 503 European samples) was 0.70 for the *FCGR3A* gene (67% reliable PSVs) but it was only 0.29 for the *SMN1* gene (15% reliable PSVs). In addition, Supplementary Fig. 8 shows that when the fraction of reliable PSVs is low, QuicK-mer2 tends to generate ambiguous *ParalogCN* values: 49% of *SMN1 ParalogCN* values from QuicK-mer2 are closer to a half-integer than to any integer.

A high frequency of unreliable PSVs is expected to adversely impact not only *ParalogCN* estimation but also short read mapping and variant calling since short-read mapping tools rely on PSVs to distinguish between different repeat copies. We used simulations to

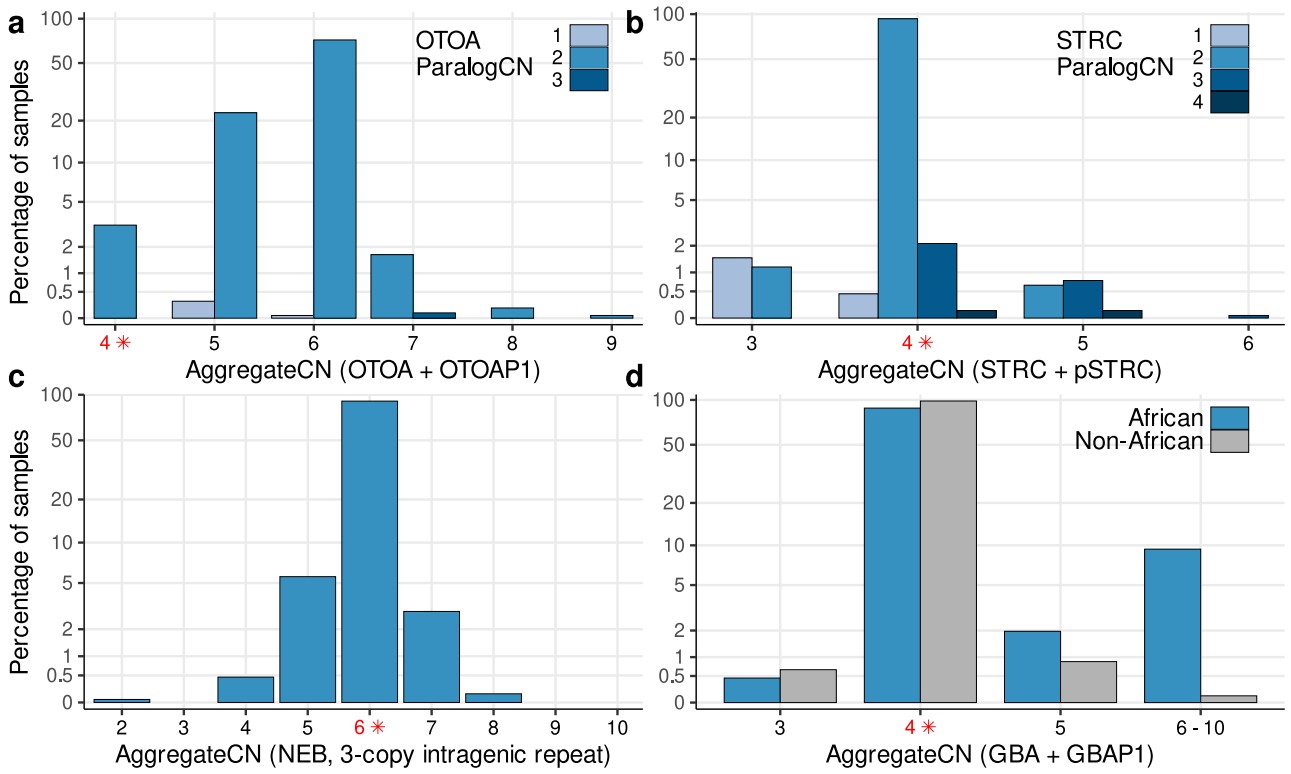

**Fig. 4 Distribution of aggregate (*AggregateCN*) and paralog-specific (*ParalogCN*) copy number values across 2504 samples from 1kGP for four disease-associated genes.** The reference copy number is shown in red and marked with an asterisk. For the *OTOA* and *STRC* loci (**a** and **b**), the *ParalogCN* distribution for each *AggregateCN* bin is also shown. **a** For the *OTOA* gene, the most frequent *AggregateCN* is 6 while the reference copy number is 4, indicating the presence of a missing repeat copy in the reference genome. **b** 1.5% samples exhibit heterozygous deletion of the *STRC* gene while no samples have a homozygous deletion. **c** For the *NEB* gene, *AggregateCN* varies between 2 and 8 across 1kGP samples while previously reported pathogenic alleles at this locus had copy number ≥9. **d** A duplication event that includes both *GBA* and *GBAP1* is frequent in African populations (9.4% of individuals have *AggregateCN* of 6–10) and almost absent in non-African populations. For completeness, the panel includes samples with *AggregateCN* quality <20.

assess the impact of the frequency of reliable PSVs on variant calling accuracy at the *SMN1/2* locus. When all PSVs were reliable, state-of-the-art variant calling tools—GATK HaplotypeCaller[37] and Freebayes[38]—achieved a recall of 0.52 and 0.55 respectively with a high precision (>0.96) for variant calling. However, when we incorporated unreliable PSVs (identified from the analysis of 1kGP data, see Methods) in the simulated reads, the precision reduced significantly to 0.56 and 0.59 and the recall decreased to 0.25 and 0.29 for the two methods.

## Discussion

In this paper, we described a new computational method (and software tool), Parascopy, specifically designed for estimation of copy number for LCRs in the human genome using WGS data. Parascopy leverages WGS data from multiple individuals to automatically account for sequencing biases and estimate aggregate and paralog-specific copy number profiles across specified region(s). Unlike some existing methods that require re-mapping or k-mer analysis of the entire WGS data, Parascopy uses a targeted approach that extracts and analyzes only reads relevant for each repeat loci from existing alignments. This allows it to efficiently estimate copy number for individual repeat loci across thousands of samples. We benchmarked Parascopy's accuracy using experimental copy number data for a number of genes and concordance analysis on replicate samples and it proved to be significantly more accurate than two existing methods—one designed for estimation of paralog-specific copy number (QuicK-mer2) and the second for genome-wide copy number variant analysis (CNVnator). Parascopy's estimates of aggregate and paralog-specific copy number

are robust to variation in sequencing biases and read depth as well as missing repeat copies in the reference genome.

A number of computational methods have been developed for detecting copy number variants from WGS data by modeling read depth[16,17,20,39]. Most of these methods are designed for analysis of unique regions of the genome and do not focus on repetitive regions of the genome. Parascopy has been developed to fill this gap and uses a two-step approach where it first estimates aggregate copy number (by aggregating reads mapped to homologous regions) and then estimates paralog-specific copy number by careful modeling of PSVs. A similar approach has been used by the SMNCopyNumberCaller method[26]—a method designed for analysis of a single duplicated gene. However, Parascopy's general framework works for any low-copy repeat in the human genome and does not make assumptions about which PSVs can be used to distinguish the paralogous repeat copies. Instead, Parascopy explicitly models and estimates population allele frequencies for each PSV using WGS data for multiple samples and is the first method to do so. Analysis of WGS data at the *SMN1/2* locus demonstrated the ability of Parascopy to correctly identify reliable PSVs and also showed that using a fixed set of PSVs for estimating *ParalogCN* can potentially result in incorrect estimates.

Analysis of PSV allele frequencies using 1000 Genomes data showed that reliable PSVs were highly consistent across populations, however, the frequency of reliable PSVs varied significantly across genes. Information about reliable PSVs is not only useful for estimating paralog-specific copy number but is also relevant for read mapping and variant calling in LCRs. We have

previously shown that post-processing of long read alignments using a probabilistic model that models genotypes for PSVs improves read mapping in LCRs[40]. It is well documented that short-read variant calling in LCR regions exhibits a higher rate of false negatives (due to low mappability) and false positives compared to unique regions of the genome[41]. Knowledge about reliable PSVs has the potential to improve short-read mapping and variant calling accuracy in such regions.

Parascopy has several limitations and avenues for further improvement. Parascopy's accuracy is lower for short regions and for regions with very high copy number (>7). Nevertheless, Parascopy was able to estimate aggregate copy number with greater accuracy for the *AMY1* locus than existing methods. In addition, it cannot estimate *ParalogCN* for loci with high reference copy number (difficult to model large number of possible paralog-specific copy number values) or loci with a very low fraction of reliable PSVs. Parascopy currently works for only WGS data, however, information about allele-specific read depth at PSVs can potentially be used to infer copy number from targeted sequencing assays. Parascopy can estimate copy number for individual genomes using pre-computed model parameters, however, sample-specific sequencing biases may reduce the accuracy of copy number estimation. Parascopy assumes that the paralog-specific copy number for each sample is constant across the analyzed region. However, gene conversion events and hybrid alleles resulting from non-allelic homologous recombination are commonly observed at LCR loci[42,43] and can result in non-uniform paralog-specific copy number. An HMM based approach can be used to model and detect such events and we plan to explore this in future work.

Unlike variants in unique regions of the genome, small sequence and copy number variants in duplicated genes are rarely analyzed in large-scale human genetic studies. Over the last few years, a number of large-scale WGS datasets for rare and common human diseases have become available[44,45] and several others are expected to be available soon[46]. We expect Parascopy to be a valuable tool for analyzing such large-scale WGS datasets to identify novel genotype-phenotype associations. In addition, copy number profiles from such datasets will be useful for prioritizing pathogenic copy number changes in duplicated genes in the human genome. Finally, Parascopy can be useful for assessing the completeness and correctness of de novo assemblies at LCRs which can be challenging to assemble correctly even using long reads.

## Methods

Given short sequence reads from WGS aligned to a reference genome for one or more samples, Parascopy jointly analyzes reads aligned to a genomic region $R$ and its homologous sequences to estimate aggregate copy number (*AggregateCN*)— number of copies of $R$ and its paralogs—as well as paralog-specific copy number (*ParalogCN*)—number of copies of each paralog. The estimation is performed jointly across all samples in two steps: (i) *AggregateCN* profiles are estimated first using read depth in fixed length windows and (ii) *ParalogCN* values are estimated using allele-specific read counts at PSVs and *AggregateCN* profiles. The workflow of the method is presented in Fig. 1a. Before copy number estimation, background read depth distributions are estimated for each sample using reads mapped to unique regions of the genome.

**Construction of homology table**. Parascopy uses a precomputed table of homologous regions in the genome (homology table) to identify the paralogous regions for a given genomic region. This homology table stores pairwise duplications in a BED format that allows for indexing and fast retrieval of all duplications overlapping a given genomic region. For each duplication, we store a sequence alignment, length, sequence similarity and other characteristics, which allow for convenient filtering of duplications. Duplications with more than two copies are identified from overlapping pairwise duplications, and PSVs are extracted from the sequence alignments. The homology table is constructed by self-alignment of the reference human genome to itself using BWA[47] (see Supplementary Methods). The table is designed to store primarily LCRs; therefore, sequences with too many (>10) pairwise alignments—that typically correspond to interspersed repeats—are discarded and the regions appropriately flagged in the table.

**Normalization of read depth**. To infer copy number from read depth, we use information from the observed read depth in a large number of non-repetitive regions of the genome (assumed to have a copy number of 2) for each sample. Briefly, read depth is calculated for windows of fixed length (default 100 bp) selected from unique regions of the genome by assigning mapped reads to the window that contains the center of the first read in each read-pair. Windows that have a high fraction (≥10%) of (i) low mapping quality reads (<10), (ii) reads not mapped in proper pairs, or (iii) soft-clipped reads—are marked as *irregular* and not used for normalization. We considered several distributions to model the read depth distribution and found that the Negative Binomial (NB) distribution provided a better fit for the read depth distribution for PCR-based WGS data compared to the Poisson distribution (see Supplementary Fig. 4 for an example). Therefore, we use the NB distribution to model the variation in read depth across windows in unique regions of the genome. To account for variation in read depth due to GC-content, we use separate NB parameters for each GC-content value (see Supplementary Methods). This procedure is performed independently for each sample and only needs to be done once for each sample independent of the number of repeat loci.

We utilize a set of genomic windows used by the SMNCopyNumberCaller tool[26] for estimating background read depth. We split the set of regions into short windows of length 100 bp. To increase the number of windows with extreme GC-content (≤35% or ≥55%), we select such windows in a 5 kb neighborhood of the original set of genomic regions. Finally, we discard all windows with a distance <500 bp to any duplication in the homology table. This procedure yields ~90,000 windows for both hg19 and hg38 versions of the human genome.

**Identifying homologous regions and calculating aggregate read depth**. For a given region $R$, we find all pairwise homologies overlapping $R$ from the homology table. To reduce complexity, we skip short duplications (<500 bp). The homologous segments are used to split $R$ into subregions or segments of constant *reference copy number*. Note that the reference copy number of a two-copy duplication is equal to 4 as the human genome is diploid. Next, for each sample, we extract reads aligned to the homologous regions and re-map these reads to the region $R$. This re-mapping is efficiently done using the precomputed alignments between $R$ and its homologous sequences that are stored in the homology table. Each subregion of constant reference copy number is divided into non-overlapping windows of fixed length and aggregate read depth is computed for each window using the reads from the region $R$ and the reads re-mapped to $R$. The read assigning procedure is same as for the background read depth analysis. For each window we calculate the fraction of reads with soft-clipping and the fraction of reads not mapped in a proper pair and filter out windows (and one flanking window on either side) if they are *irregular* in more than 10% of the samples.

For some loci, two subregions with the same reference copy number are interrupted by a short region with a different reference copy number (for example by an interspersed repeat). We group such subregions (if they are separated by <2000 bp) into *region groups* and all subsequent analysis is performed independently for each region group.

**Estimating aggregate copy number profiles jointly for multiple samples**. To estimate the aggregate copy number (*AggregateCN*) profiles, we construct a HMM[48] for each region group. For a single sample $s$, the aggregate read depth values in the windows across the region represent the observed values and the *AggregateCN* value for each window forms a set of hidden states. For each window, we consider $K$ possible *AggregateCN* values where $K$ is selected based on the reference copy number for the region and observed aggregate read depth for all samples (see Supplementary Methods).

To reduce the number of parameters, we define all transition probabilities based on two parameters for each consecutive pair of windows: $\tilde{a}_{\nearrow w}, \tilde{a}_{\searrow w} \in \left[0, \frac{1}{10}\right]$. We define a jump from *AggregateCN* $i$ to a larger *AggregateCN* $j$ as $a_{ijw} = \tilde{a}_{\nearrow w}^{j-i}$ and to a smaller *AggregateCN* $k$ as $a_{ikw} = \tilde{a}_{\searrow w}^{i-k}$. The transition probability $a_{iiw}$ therefore equals $1 - \sum_{j \neq i} a_{ijw}$. By default, both $\tilde{a}_{\nearrow w}$ and $\tilde{a}_{\searrow w}$ are set to $10^{-5}$ on the first iteration, and the initial state distribution is set to $\pi_{\neg\text{ref}} = \frac{1}{|S|}$ for all non-reference copy number states and $\pi_{\text{ref}} = 1 - \frac{K}{|S|}$ for the reference copy number.

Let $o_w^{(s)}$ be the aggregate read depth for sample $s$ in window $w$ and let $n_w^{(s)}$ and $p_w^{(s)}$ be the parameters of the NB distribution corresponding to the GC-content of window $w$ (estimated separately for each sample). The emission probability for copy number $c$ (hidden state) and window $w$ is defined as:

$$b_w^{(s)}(c) = P_{\text{NB}}\left(o_w^{(s)}; n_w^{(s)} \cdot c/2, p_w^{(s)}\right).$$

For each sample, we use the Forward-Backward algorithm[49] to obtain $\gamma_{c,w}^{(s)}$—the probability that sample $s$ has copy number $c$ in window $w$. Next, HMM parameters—emission and transition probabilities as well as the initial state distribution—are updated iteratively using $\gamma_{c,w}^{(s)}$. The emission probabilities are updated using a scale parameter $m_w$ for each window $w$ that models window-specific biases in sequencing read depth that are not captured by GC-content based modeling[19]. In

the first iteration, $m_w$ is initialized to 1 for all windows. This parameter scales the expected read depth for each window (equally for all samples) to be higher or lower than the default value and is estimated using a maximum likelihood procedure and the $\gamma_{c,w}^{(s)}$ estimates (see Supplementary Methods and Supplementary Fig. 5).

To update initial and transition probabilities, we use a procedure similar to Baum–Welch algorithm[48] (see Supplementary Methods). The intuition underlying these updates is that the initial state probabilities correspond to population frequencies of the aggregate copy number values at the start of the region, while shared deletion or duplication events result in increased transition probabilities between the states of adjacent windows where the events happen. This iterative procedure—Forward-Backward algorithm for each sample followed by joint update of the HMM parameters—is run until the log-likelihood of the data (summed over all samples) converges (See Supplementary Methods). Finally, we run the Viterbi algorithm[50] to find most probable *AggregateCN* profile for each sample.

**Estimating paralog-specific copy number using PSVs.** Once the *AggregateCN* profiles are estimated for each sample, we estimate paralog-specific copy number (*ParalogCN*) values using the allele-specific read counts at PSV sites across the region $R$. For a region with reference copy number $c_r$ and a sample $s$ with *AggregateCN* $c_s$, the paralog-specific copy number is defined as an integer tuple of length $c_r/2$ that sums up to $c_s$. Each tuple element represents the copy number of a specific copy of the duplication and order of the copies is the same for all samples. Note that there are $\binom{3/2 \cdot c_r - 1}{c_r}$ possible paralog-specific copy number tuples for each sample. We assume that the *ParalogCN* does not change in a region with constant *AggregateCN*.

PSVs are defined based on the reference genome assembly and only those PSVs for which the allele on a specific paralog of a duplication is invariant in the population are useful for estimating *ParalogCN*. Since some PSVs are known to correspond to polymorphisms, we model the frequency of the reference allele at a PSV site $v$ and paralog $k$ as $f_{vk}$ where $f_{vk} \in [0, 1]$. We call a PSV $v$ *reliable* if all its $f$ values are close to 1: $\min_{k=1}^{c_r/2} f_{vk} \geq 0.95$. Such PSVs can be used as markers of each paralog and are useful for estimating *ParalogCN*.

Given sequence data from multiple samples, we want to estimate two sets of variables: (i) *ParalogCN* for each sample, and (ii) PSV frequency matrix $f$ where $f_{vk}$ is the frequency of the reference allele for PSV $v$ on the $k$-th copy. We use an Expectation-Maximization (EM) algorithm to solve this problem where sample *ParalogCN* values are hidden variables and the matrix $f$ is an unknown parameter (see Supplementary Methods for details). In order to reduce computational complexity, we apply the EM algorithm only to those samples for which *AggregateCN* is equal to the reference copy number $c_r$ (minimum of 50 samples). Once the PSV $f$ values are determined, we calculate the *ParalogCN* for all samples individually. For this, we run the E-step of the EM algorithm using only reliable PSVs. Parascopy does not estimate *ParalogCN* for loci with very high *AggregateCN* or reference copy number (>8) and for loci with very high number of possible *ParalogCN* tuples (>500) to limit run-time.

**Estimating copy number values for a single individual.** Parascopy is designed to estimate *AggregateCN* and *ParalogCN* profiles using data for multiple samples, therefore, analyzing a single sample or a small number of samples may not produce very accurate results, particularly for *ParalogCN*. To enable the analysis of individual samples, we can use model parameters estimated from a population of samples, e.g., 1000 Genomes Project. Model parameters include initial and transition probabilities of the aggregate copy number HMM, as well as a set of window-specific scale parameters $\{m_w\}$ and PSV frequency matrix $f$. This allows us to quickly analyze individual samples using precomputed model parameters for multiple duplicated loci.

**Genome-wide set of duplicated gene loci.** To obtain a set of duplicated gene loci, we started with a set of 1168 duplicated genes that were reported previously[3] as having at least one exon that is difficult to map using short reads due to high sequence similarity to one or more other loci. From these, we removed 124 genes that are known to vary extensively in copy number[24] and 88 genes that are missing from the GENCODE annotation v37[51]. In addition, we discarded 257 genes that did not overlap any duplication longer than 2 kilobases in the homology table. The remaining genes were merged into 564 loci, which were then manually filtered in order to remove high copy number regions and regions with complex duplication structures. For some loci, we included additional flanking sequence to provide more useful information for copy number detection. The final set of duplicated gene loci set contained 167 regions, with all homologous regions covering 12.6 Mb.

**Analysis of 1000 Genomes samples at 167 repeat loci using Parascopy.** We used high-coverage (30×) whole-genome sequence data for 2504 samples from the 1000 Genomes Project (1kGP) generated by the New York Genome Center[29]. All samples were sequenced using a PCR-free library preparation protocol and cram files aligned to the reference human genome hg38 were used for analysis. The 2504 samples were divided into 5 groups based according to their continental population and each group of samples was analyzed together in a single run of Parascopy. To assess copy number concordance in trios, we analyzed WGS

data from an additional set of 698 related samples from the 1kGP resource. The 698 samples were analyzed independently of the 2504 samples in a single run of Parascopy.

**Copy number benchmarking using experimental data.** For a given duplicated locus, Parascopy outputs integer *AggregateCN* and *ParalogCN* estimates for various subregions of the locus. In contrast, QuicK-mer2 and CNVnator output fractional *ParalogCN* values for various subregions throughout the whole genome and do not output *AggregateCN* directly. In order to facilitate a direct comparison, we extract Parascopy, QuicK-mer2 and CNVnator copy number estimates that overlap single positions within multiple copies of 167 duplicated loci (Supplementary Data 7). As CNVnator does not output *ParalogCN* estimates in the absence of deletions and duplications, we treat missing CNVnator values as *ParalogCN* = 2. Next, for each locus, we sum fractional *ParalogCN* values and round the sum to the nearest integer to obtain *AggregateCN* estimate; In addition, we round every fractional *ParalogCN* value to obtain integer *ParalogCN* estimates.

Throughout the paper, we consider Parascopy copy number values to have high quality if their Phred-score is at least 20. Likewise, we convert CNVnator $E$ values into Phred-scores and say that the *AggregateCN* and *ParalogCN* estimates have high quality if the scores are ≥20 across all copies. We assume that all QuicK-mer2 copy number estimates have high quality, as the method does not output any quality measures.

To measure copy number estimation accuracy, we compare Parascopy, QuicK-mer2 and CNVnator *AggregateCN* and *ParalogCN* values derived from WGS data against corresponding copy number estimates based on experimental observations (Tables 1 and 2, Supplementary Table 1 and Supplementary Data 2) for the same locus and the same individuals. If the experimental copy number observation is a fractional number, we round it to the nearest integer. Copy number estimate is *correct* if it matches completely with the experimental observation for the same sample. In Tables 1 and 2, QuicK-mer2 *AggregateCN* and *ParalogCN* were aggregated across the duplicated genes, and median copy number value was selected. This procedure improved QuicK-mer2 accuracy; however, we did not perform it for all 167 duplicated loci, as it requires a careful case-by-case approach, especially in complex duplications.

SMNCopyNumberCaller[26] v1.1.1, QuicK-mer2[25] build 2021 and CNVnator[17] v0.4.1 were run on the WGS datasets using default parameters. In addition, QuicK-mer2 copy number estimates for 2457 1kGP samples were downloaded from https://github.com/KiddLab/kmer_1KG.

**Assessing consistency of copy number estimates.** To evaluate Parascopy, QuicK-mer2 and CNVnator robustness, we compare *AggregateCN* and *ParalogCN* estimates obtained for the same individuals based on two independent WGS datasets for 83 Han Chinese samples: PCR-free IGSR[32] dataset and PCR-based BGI[31] dataset. Copy number estimates were selected based on a set of positions within 167 duplicated loci in both datasets; in this way each sample is associated with 167 pairs of *AggregateCN* and *ParalogCN* values. A pair of copy number estimates is considered *available*, if the corresponding copy number estimates have high quality in both datasets. Accordingly, a pair of copy number estimates is *concordant*, if it is available and the corresponding copy number estimates match completely.

To assess robustness of Parascopy copy number estimates to variation in read depth and model parameters, we create two sets of Parascopy model parameters using 503 and 504 samples from the European and East-Asian continental populations, respectively. We analyze the same set of samples (103 Han Chinese samples and 107 Iberian samples) using both sets of model parameters and evaluate the concordance of resulting *AggregateCN* and *ParalogCN* values. In addition, we subsample the 107 Iberian samples to one-third and two-third coverage, analyze subsampled datasets using East-Asian model parameters, and compare resulting copy number estimates against those obtained using full-coverage dataset and European model parameters.

**Paralog-specific copy number validation using trios.** In order to assess the accuracy of paralog-specific copy number estimates using Parascopy, we analyzed 602 trios with WGS data from the extended 1kGP[32]. The child in each trio was analyzed independently from the two parents to avoid any bias (except for 9 trios that consisted entirely of IGSR relatives). To assess consistency of *ParalogCN* values in trios, we modeled the population frequencies of paralog-specific copy number for a single chromosome using the observed diploid observations (see Supplementary Methods). For each trio with high quality (≥20) *ParalogCN* estimates, we calculated the probability of observing the child's *ParalogCN* given the *ParalogCN* estimates of both parents. A trio was considered *discordant* if this probability was <0.01.

**Measuring the effect of unreliable PSVs on variant calling.** In order to evaluate the consequences of unreliable PSVs ($f < 0.95$) on variant calling, we used the NEAT short-read simulation tool[52] v3.0 to generate a baseline set of single nucleotide variants (SNVs, ≈ 1 SNV per 1 kb) and high-coverage (30×) WGS data with and without unreliable PSVs. To introduce unreliable PSVs we randomly replaced PSV reference alleles with alleles from another copy of the duplication according to the frequencies of PSV reference alleles ($f$ values) in *SMN1/2* locus in 503 European samples from the 1kGP. This procedure yielded 31 homozygous and 33

heterozygous SNVs corresponding to unreliable PSVs. Next, we called variants in both copies of the duplication (total length 111 kb) using GATK HaplotypeCaller[37] v4.2.2 and Freebayes[38] v1.3.5 and compared results with baseline sets of SNVs using RTG tools[53] v3.12.1.

**Reporting summary**. Further information on research design is available in the Nature Research Reporting Summary linked to this article.

## Data availability

The analyses presented in this paper are based on the high-coverage whole genome sequencing data of 1000 Genomes Project samples that was generated at the New York Genome Center with funds provided by NHGRI Grant 3UM1HG008901-03S1. This sequencing data is available via ENA Study PRJEB31736 and PRJEB36890. The whole-genome sequence data for 90 Han Chinese samples is available from ENA Study PRJEB11005. For this dataset, we used aligned reads downloaded from https://ftp.1000genomes.ebi.ac.uk/vol1/ftp/data_collections/han_chinese_high_coverage for analysis. Source data are provided with this paper.

## Code availability

Parascopy is implemented in the Python programming language and is freely available for download at https://github.com/tprodanov/parascopy[54]. It is also available via conda (conda install -c bioconda parascopy). Parascopy requires BAM/CRAM files for one or more samples, a reference genome sequence and a homology table (provided for human reference genomes hg19 and hg38) as input.

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

## Acknowledgements

We thank Vineet Bafna and Melissa Gymrek for helpful discussions and feedback on the computational methods described here. We would also like to thank Raymon Vijzelaar for sharing updated MLPA copy number data for the *SMN1* locus. This work was supported by a grant from the National Human Genome Research Institute (NHGRI; R01 HG010149).

## Author contributions

V.B. conceived the project; T.P. and V.B. designed the algorithm; T.P. implemented the algorithm; T.P. and V.B. performed the analyses; V.B. and T.P. wrote the paper.

## Competing interests

The authors declare no competing interests.
