## [Peer Review File · Nature Communications]

Title: Robust and accurate estimation of paralog-specific copy number for duplicated genes using whole-genome sequencingREVIEWER COMMENTS

Reviewer #1 (Remarks to the Author):

The authors present a tool for estimating copy number for repetitive regions of the genome that are typically ignored because they are difficult. This is an important problem and innovations in this area would likely have an impact on medical genetics.

I am not asking the authors to perform this analysis as part of this paper, but I would really like to know how well this could work with exome sequencing. The authors show that Parascopy is robust to a number of different copy number and library differences, and the impact of the paper would be much higher this extends to handle the probe variability. If the authors already know that it only works for whole genome sequencing, then it may be worth add that statement so someone uses their methods on exome data.

My biggest concern with the paper is the lack of a comprehensive comparison to other similar methods. The standard for publishing a novel genomic method is high and it requires a clear and convincing set of experiments that demonstrate the relative improvement of the proposed algorithm in terms of performance, sensitivity, and specificity.

The authors need to justify a specialized copy number tool. GenomeStrip was specially called out in the text as a method that works in duplicated regions. Presumably it does not work well enough in the regions that Parascopy targets, but those experiments must be done and those data must be shown.

As far as duplication-specific comparisons, the authors focus on a handful of encouraging results that indicate Parascopy's sensitivity, which feels cherry picked. There is also no mention of false discovery rates. The authors need to include a more comprehensive comparison and need to include more methods in that comparison. I would suggest setting up a head-to-head comparison of all 167 regions that Parascopy targets for the methods mentioned on page 2.

Since Parascopy is a targeted method, the authors should also quantify false-negatives vs a genome-wide method.

Most of these results are given in Table 1, which I found confusing. Method papers typically give figures that show true positive / false positive tradeoff curves for several methods. Such a figure would also give the authors the opportunity to explore the sensitivity of their 0.95 f-value threshold for PSV reliability. Perhaps relaxing this value would increase sensitivity.

The other issue I had was with reliability. PSV, psCN, agCN, CN. There are just too many of these for me to easily read and digest the results. I would encourage the sacrifice space in name of clarity.

Reviewer #2 (Remarks to the Author):

Prodanov and Bansal present a method, Parascopy, to call copy number variants of duplicated regions using a Hidden Markov Model and paralog-specific variants. The manuscript is written very clearly and the results are promising. The robustness compared to CHM13 T2T is important.

While this problem has been studied for quite some time, the ideas presented can, in aggregate, help advance the field of CNV discovery using short reads. In particular, a careful approach to first estimate the total copy number of a gene, and then discover the paralog-specific copies is shown to be effective.

MAJOR-1

The discovery of aggregate copy number of duplicated copies is not novel, and forms the basis of the approach used in citation 23 (Alkan et al). However, the use of the homology table in lieu of multi-mapping of reads should speed the calculation up. A major question is: how dependent is this on the annotation of duplicate copies of regions, particularly in complex mosaic duplications? A straightforward way to answer this is to compare the estimate of duplication copies (agCN) using the duplication table to that derived from mrsFAST mappings. Since these mappings are slow, a handful of samples is sufficient. Alternatively, one can track supplementary alignments produced by bwa to count aggregate read depth over duplications.

MAJOR-2

The measure of Mendelian inheritance shows high accuracy, but the number of duplications that are reference allele are also often high. It would be good to state, for the number of children with an expansion, the percentage of parents with copy numbers compatible with the child.

MAJOR-3

There should either be a larger discussion of how previous methods for CNV discovery do not work well for low copy repeats, or a direct comparison of parascopy against them. An example of callers is here: <https://www.sciencedirect.com/science/article/pii/S1532046419300929#f0005> .

MINOR-1

The method could be powerful at detecting gene conversions. These will show up as adjacent PSVs that flip gene copy assignments.

MINOR-2

The homology table may be constructed only mapping genes that are profiled, not the whole genome alignments. This will make the problem of interspersed repeats less of an issue, though the method seems to work fine.

We thank the reviewers for the useful comments and the suggestions for improving the manuscript. We have performed many additional evaluations to address the reviewers' concerns. We have significantly revised our manuscript to incorporate the new results (changed highlighted in red). Our point-by-point response to these comments is included below.

Major updates to the Tables and Figures:

- **Main text:**
 - We have removed Table 1 from the original manuscript and replaced it with revised Tables 1 & 2 that benchmark the different methods on aggregate copy number and paralog-specific copy number respectively.
 - We have added Table 3 with results for benchmarking the concordance of different methods on 83 replicate samples
- **Supplementary data:**
 - Supplementary Figure 1 from the original manuscript has been removed and replaced with a figure comparing Parascopy and experimental copy number estimates for the NP4YR/2 duplication
 - Supplementary Figure 2 has been revised to compare the performance of the three different methods at the AMY1A/B/C locus
 - Supplementary Figure 5 from the original manuscript has been removed and replaced by Supplementary Table 2 for consistency with Table 3 in the main text
 - Supplementary Figures 3 and 9 have been newly added
 - Supplementary Tables 1 and 2 have been newly added

REVIEWER COMMENTS

Reviewer #1 (Remarks to the Author):

The authors present a tool for estimating copy number for repetitive regions of the genome that are typically ignored because they are difficult. This is an important problem and innovations in this area would likely have an impact on medical genetics.

I am not asking the authors to perform this analysis as part of this paper, but I would really like to know how well this could work with exome sequencing. The authors show that Parascopy is robust to a number of different copy number and library differences, and the impact of the paper would be much higher this extends to handle the probe variability. If the authors already know that it only works for whole genome sequencing, then it may be worth add that statement so someone uses their methods on exome data.

Response: We agree with the reviewer's comment that it would be valuable if our method worked for exome data as well. However, the method and the current implementation is

designed to work with WGS data only. We note that almost all methods for CNV analysis from WGS data don't work well for exome data. In principle, the method could work with exome data but demonstrating this will require substantial work (normalization of copy number and validation of the results). In addition, estimation of paralog-specific copy number relies on PSVs which are significantly less in number in exome data compared to WGS data. We have added a statement to the github page of Parascopy stating that the method does not work for exome or targeted sequencing data.

My biggest concern with the paper is the lack of a comprehensive comparison to other similar methods. The standard for publishing a novel genomic method is high and it requires a clear and convincing set of experiments that demonstrate the relative improvement of the proposed algorithm in terms of performance, sensitivity, and specificity.

Response: We appreciate the concern of the reviewer and agree that additional benchmarking is needed. To address this, we have included the CNVnator method - a state-of-the-art tool for copy number analysis from read depth in WGS data - for benchmarking. Although CNVnator is not designed specifically for copy number analysis of repeat regions, it is one of the most widely used tools for CNV analysis. We have compared the performance of the three methods for estimating aggregate copy number, paralog-specific copy number and boundaries of copy number changes within a gene using experimental data and concordance analysis for replicate samples. These comparisons are summarized below:

- 1) In Tables 1 and 2 of the revised manuscript, we compare Parascopy with Quick-mer2 and CNVnator in terms of accuracy of aggregate and paralog-specific copy number using all experimental copy number datasets that we could find in the literature for duplicated genes and 1000 Genomes samples. We find that the copy number estimates obtained using Parascopy are more accurate (or equally accurate) than those obtained using the two other methods.
- 2) In Table 3 (newly added), we now compare our method with Quick-mer2 and CNVnator by assessing the concordance (of copy number values) between pairs of replicate samples across 167 duplicated loci. The replicate samples correspond to 83 Han Chinese samples with WGS data derived from two different library preparation methods. We find that Parascopy's copy number estimates - aggregate and paralog-specific - are much more concordant (5-10 times lower mean absolute difference) than other methods. For Parascopy, 98.7% of the *aggregateCN* values (quality ≥ 20) are concordant between replicate samples compared to 74.9% and 86.9% for Quick-mer2 and CNVnator respectively. Similar for *paralogCN* values, Parascopy had a concordance rate of 99.8% compared to 81.1% and 85.5% for Quick-mer2 and CNVnator.
- 3) We compared the performance of the different methods for identifying the boundaries of copy number changes within a gene using data for two LCR loci: SMN1/2 and PMS2/PMS2CL. The results are shown in Supplementary Table 1 and Supplementary Figure 3. Parascopy achieves higher sensitivity and specificity for detecting these local copy number changes than both Quick-mer2 and CNVnator.

Overall, these comparisons clearly demonstrate the significantly greater accuracy of Parascopy for estimating aggregate and paralog-specific copy number.

The authors need to justify a specialized copy number tool. GenomeStrip was specially called out in the text as a method that works in duplicated regions. Presumably it does not work well enough in the regions that Parascopy targets, but those experiments must be done and those data must be shown.

Response: We have now included a comparison with CNVnator, a widely used tool for CNV analysis from WGS data that is not specialized for duplicated regions. Benchmarking using experimental copy number data (Tables 1 and 2) shows that Parascopy outperforms both CNVnator and Quick-mer2. Assessment of concordance between replicate samples shows that Parascopy's mean absolute difference is significantly lower than the other two methods (Table 3). Handsaker et al. (Nat. Gen 2015) described the estimation and analysis of copy number using GenomeStrip in duplicated regions of the genome. However, the tool – as available on the GenomeStrip website – is limited to detecting CNVs in unique regions and does not provide any options for doing copy number analysis of repeat regions. We have attempted to contact the authors by email but did not receive a response. Therefore, we did not include this tool in the benchmarking.

As far as duplication-specific comparisons, the authors focus on a handful of encouraging results that indicate Parascopy's sensitivity, which feels cherry picked. There is also no mention of false discovery rates. The authors need to include a more comprehensive comparison and need to include more methods in that comparison. I would suggest setting up a head-to-head comparison of all 167 regions that Parascopy targets for the methods mentioned on page 2.

Response: The genes and loci included in Tables 1 and 2 correspond to all loci for which we could find experimental copy number data (on 1000 Genomes samples) from the literature. Parascopy is a targeted method that can detect locus-wide changes in copy number as well as partial changes in copy number. In the absence of more extensive experimental data, we have used concordance analysis for pairs of 83 replicate samples to perform a comprehensive comparison of Parascopy with two previously published methods (Quick-mer2 and CNVnator). The comparison has been performed using the genome-wide set of 167 regions that Parascopy targets.

Since Parascopy is a targeted method, the authors should also quantify false-negatives vs a genome-wide method.

Response: Parascopy is designed to analyze individual repeat loci which makes it challenging to compare its performance with genome-wide methods such as Quick-mer2 and CNVnator. We have compared its accuracy with these two methods for individual loci. The accuracy numbers reported in Tables 1 and 2 correspond to the percentage of samples at each locus for

which the WGS-based copy number matched the experimental copy number. This includes both false-positives (incorrectly detected copy number change) and false-negatives (copy number change relative to the reference not detected). Many of the loci reported in Tables 1 and 2 have significant variation in copy number across samples. In addition, we have evaluated sensitivity and specificity for detection of partial copy number changes in duplicated genes SMN1/2 and PMS2/CL (See SupplementaryTable 1).

Most of these results are given in Table 1, which I found confusing. Method papers typically give figures that show true positive / false positive tradeoff curves for several methods. Such a figure would also give the authors the opportunity to explore the sensitivity of their 0.95 f-value threshold for PSV reliability. Perhaps relaxing this value would increase sensitivity.

Response: We have reorganized Table 1 for clarity and split the original table into two tables: Table 1 for benchmarking of aggregate copy number and Table 2 for paralog-specific copy number. The accuracy number reported in Tables 1 and 2 for each method corresponds to the percentage of copy number estimates from WGS data (for each gene) that matched the experimental copy number value for the same sample. We have also added information about the fraction of copy number values that are not reported (missing) for applicable loci in Tables 1 and 2. In Supp.Table 1, we report the sensitivity and specificity of detecting partial deletion events (for two genes) for each of the three methods. We evaluated the impact of the f -value threshold on the estimation of paralog-specific copy number for the 4 loci from Table 2 by considering different thresholds (0.85, 0.9, 0.95, 0.99). The results showed that the paralog-specific copy number accuracy was unchanged.

The other issue I had was with reliability. PSV, psCN, agCN, CN. There are just too many of these for me to easily read and digest the results. I would encourage the sacrifice space in name of clarity.

Response: We have updated the manuscript to remove the abbreviations CN, agCN and psCN. We now use the terms 'ParalogCN' and 'AggregateCN' to refer to paralog-specific copy number and aggregate copy number respectively. Similarly, we have revised the manuscript to only use the terms reliable and unreliable PSVs to denote PSVs with f -values ≥ 0.95 or less than 0.95 respectively.

Reviewer #2 (Remarks to the Author):

Prodanov and Bansal present a method, Parascopy, to call copy number variants of duplicated regions using a Hidden Markov Model and paralog-specific variants. The manuscript is written very clearly and the results are promising. The robustness compared to CHM13 T2T is important.

While this problem has been studied for quite some time, the ideas presented can, in aggregate, help advance the field of CNV discovery using short reads. In particular, a careful approach to

first estimate the total copy number of a gene, and then discover the paralog-specific copies is shown to be effective.

Response: We thank the reviewer for appreciating our work.

MAJOR-1

The discovery of aggregate copy number of duplicated copies is not novel, and forms the basis of the approach used in citation 23 (Alkan et al). However, the use of the homology table in lieu of multi-mapping of reads should speed the calculation up. A major question is: how dependent is this on the annotation of duplicate copies of regions, particularly in complex mosaic duplications? A straightforward way to answer this is to compare the estimate of duplication copies (agCN) using the duplication table to that derived from mrsFAST mappings. Since these mappings are slow, a handful of samples is sufficient. Alternatively, one can track supplementary alignments produced by bwa to count aggregate read depth over duplications.

Response: The homology table is constructed using the entire reference genome and does not rely on annotation of duplications in the genome. It can capture simple as well as complex duplications. The homology table allows us to generate all mapping locations of multi-mapped reads for an individual low-copy repeat locus efficiently. This works for any mapping tool and does not require re-mapping all reads. We have tried to use BWA supplementary alignments for tracking aggregate read depth for the SMN1/2 locus; however, it resulted in under-estimating the copy number compared to the experimental data. This is because regions with moderate divergence can result in no supplementary alignments. For example a 10-bp difference can make one of the alignments much better than the other, so the second alignment will not be considered as supplementary. Using mrsFAST is also impractical for targeted copy number analysis since it requires converting the BAM/CRAM file to a FASTQ file and mapping all reads to the reference. We did try to run mrsFAST on a single sample but the run did not finish within a reasonable time-frame (96 hours).

MAJOR-2

The measure of Mendelian inheritance shows high accuracy, but the number of duplications that are reference allele are also often high. It would be good to state, for the number of children with an expansion, the percentage of parents with copy numbers compatible with the child.

Response: We thank the reviewer for this suggestion. We now report the trio concordance rate separately for trios where the child's paralog-specific copy number was greater than 2 (reference copy number). The concordance rate for such cases (95.5%, 4776/5093) was lower than the overall rate.

MAJOR-3

There should either be a larger discussion of how previous methods for CNV discovery do not work well for low copy repeats, or a direct comparison of parascopy against them. An example of callers is here: <https://www.sciencedirect.com/science/article/pii/S1532046419300929#f0005>

Response: We have included a widely used tool for CNV discovery from WGS data, CNVnator, for comparison with Parascopy (please see response to comment #1 from Reviewer 1). In addition, we have added a paragraph to the discussion highlighting the motivation for developing Parascopy and how it differs from CNV calling methods.

MINOR-1

The method could be powerful at detecting gene conversions. These will show up as adjacent PSVs that flip gene copy assignments.

Response: We thank the reviewer for this suggestion. Indeed, a simple HMM based procedure can be used to identify potential gene conversion events as well as hybrid alleles resulting from non-allelic homologous recombination. In future work, we expect to analyze the ability to detect such variants using comparison to long-read sequence data along with variant calling. We have added the following text to the discussion to reflect this point:

“Parascopy assumes that the paralog-specific copy number for each sample is constant across the analyzed region. However, gene conversion events and hybrid alleles resulting from non-allelic homologous recombination are commonly observed at LCR loci and can result in non-uniform paralog-specific copy number. An HMM based approach can be used to model and detect such events and we plan to explore this in future work.”

MINOR-2

The homology table may be constructed only mapping genes that are profiled, not the whole genome alignments. This will make the problem of interspersed repeats less of an issue, though the method seems to work fine.

Response: We thank the reviewer for this suggestion. Indeed, interspersed repeats can result in small sub-regions with high copy number within long repeats. Such regions are marked in the homology table and not used for estimating aggregate or paralog-specific copy number. We have found that this does not affect the estimation in any significant way.

REVIEWERS' COMMENTS

Reviewer #1 (Remarks to the Author):

The authors have address all of my concerns.

Reviewer #2 (Remarks to the Author):

The authors have addressed my concerns.

We thank the reviewers for their feedback. There were no specific comments that needed to be addressed.

REVIEWERS' COMMENTS

Reviewer #1 (Remarks to the Author):

The authors have address all of my concerns.

Reviewer #2 (Remarks to the Author):

The authors have addressed my concerns.